# EFFICIENT DIFFERENTIABLE APPROXIMATION OF THE GENERALIZED LOW-RANK REGULARIZATION

## ABSTRACT

Low-rank regularization (LRR) has been widely applied in various machine learning tasks, but the associated optimization is challenging. Directly optimizing the rank function under constraints is NP-hard in general. To overcome this difficulty, various relaxations of the rank function were studied. However, optimization of these relaxed LRRs typically depends on singular value decomposition, which is a time-consuming and nondifferentiable operator that cannot be optimized with gradient-based techniques. To address these challenges, in this paper we propose an efficient differentiable approximation of the generalized LRR. The considered LRR form subsumes many popular choices like the nuclear norm, the Schatten-$p$ norm, and various nonconvex relaxations. Our method enables LRR terms to be appended to loss functions in a plug-and-play fashion, and be conveniently optimized by off-the-shelf machine learning libraries. Furthermore, the proposed approximation solely depends on matrix multiplication, which is a GPU-friendly operation that enables efficient parallel implementation. In the experimental study, the proposed method is applied to various tasks, which demonstrates its versatility and efficiency.

## 1 INTRODUCTION

Low-rank structures have proven to be effective in a wide range of machine learning tasks, encompassing computer vision (Ren et al., 2022; Zhang et al., 2021; Chen et al., 2022), model compression (Idelbayev & Carreira-Perpinán, 2020; Xu et al., 2021), representation learning (Fu et al., 2021; 2022), deep neural networks (Lezama et al., 2018; Geng et al., 2021), and large language model adaptation (Hu et al., 2022a). To discover and utilize the low-rank structures, a typical paradigm is to introduce low-rank regularization (LRR) terms to the models, which can conveniently express various low-rank priors and assumptions.

However, with the presence of LRR terms, the optimization problem can be extremely difficult. It is well-known that even under linear constraints, optimizing the rank function is NP-hard (Wright & Ma, 2022). To alleviate this computational intractability, many relaxations of the rank function have been proposed. For example, the nuclear norm is known to be the tightest convex approximation of the rank function, and thus has been extensively investigated and applied (Fazel et al., 2001; Yang et al., 2016). The Schatten-$p$ norm and its variants are considered as the generalization of the nuclear norm, and also found successful applications (Xu et al., 2017; Chen et al., 2021). These relaxation techniques penalize all singular values of the target matrix simultaneously, while in practical applications it is desired that the large singular values are less impacted, so the key information of the target matrix is preserved. With this motivation, recent studies proposed novel nonconvex relaxations (Kang et al., 2015; Peng et al., 2015; Friedman, 2012; Gao et al., 2011).

Though these relaxation techniques make the regularized optimization problem solvable in principle, there are still huge obstacles in developing concrete optimization strategies. Current optimization methods can be broadly categorized as the matrix factorization approach and the rank function optimization approach, but both of them suffer severe shortcomings in practice.

The basic idea of matrix factorization is to decompose the target matrix into the product of multiple low-rank sub-matrices. This formulation allows gradient to be propagated, and so the optimization is straightforward. A key problem of this approach is that it demands strong prior knowledge about the matrix's true rank. In almost all practical situations such knowledge is unavailable, and so the

good performance frequently depends on laborious parameter selection. Arguably, the goal of rank regularization is to discover the true rank of a matrix, but this approach leaves this burden to the users.

In contrast, the rank function optimization approach incorporates a regularization term into the loss function, allowing the appropriate matrix rank to be automatically determined. However, solving the associated optimization problem is nontrivial. Previous works attempted to adapt techniques from the convex optimization literature, such as the proximal gradient (Ji & Ye, 2009; Yao et al., 2018) and alternating direction method of multipliers (ADMM) (Boyd et al., 2011) algorithm. Unfortunately, many of these methods require the loss function to be convex, which severely limits their applicability. Additionally, these approaches are inconvenient to implement, and each work needs to laboriously derive the specific optimization rules. Critically, all these methods rely on singular value decomposition (SVD). Besides being time-consuming, it is a nondifferentiable operation that hinders gradient propagation. Consequently, applying gradient-based optimization techniques or utilizing popular machine learning libraries like PyTorch (Paszke et al., 2019) and TensorFlow (Abadi et al., 2016) becomes infeasible, and the high-performing GPUs cannot be fully utilized..

In this paper, we propose a novel differentiable approximation of a general form of LRR, which covers a broad range of relaxations of the rank function. The main idea of our work is to introduce an equivalent stochastic definition of the rank function, as well as its relaxed variants. This significant discovery enables us to approximate the LRR term with finite samples and a partial sum of the series expansion. Our implementation is publicly available at github.com/xxx. The main contributions of this paper are summarized as follows:

1. We propose an efficient differentiable approximation of the generalized LRR.

2. The particular form of the LRR considered in the paper is quite general, which covers the nuclear norm, the Schatten-$p$ norm, and many recently proposed nonconvex relaxations of the rank function.

3. The proposed regularization term can be conveniently appended to a loss function in a plug-and-play fashion, and then optimized by existing machine learning libraries. In most cases, a few lines of code are sufficient to adapt our method to new problems.

4. The proposed approximation solely depends on matrix multiplication, which enables highly efficient parallel implementation.

5. We performed extensive experiments over various tasks with low-rank structures, including matrix completion, video fore-background separation, and image denoising, which demonstrate the versatility and efficiency of our proposed method.

## 2 RELATED WORK

### 2.1 RELAXATIONS OF THE RANK FUNCTION

Minimizing the rank function is generally very difficult. For example, even finding the optimal solution under the linear constraint is NP-hard (Wright & Ma, 2022). To address this challenge, various relaxations of the rank function have been proposed. The nuclear norm of a matrix $\mathbf{S}$, *i.e.*, $\|\mathbf{S}\|_* = \sum_{i=1}^r \sigma_i(\mathbf{S})$ is the tightest convex relaxation of the rank function, and is one of the most popular substitution (Fazel et al., 2001; Yang et al., 2016). However, the nuclear norm penalizes all singular values simultaneously. Since for many matrices in practical applications, the major information is captured by a few singular values, so it is desired that they are less impacted when reducing the rank. Motivated by this, advanced relaxations of the general form $\mathcal{R}(\mathbf{S}) = \sum_{i=1}^r h(\sigma_i(\mathbf{S}))$ are widely considered, where $h$ is a function that increases penalties on small singular values. Typical examples of the relaxed rank function include the $\gamma$-nuclear norm (Kang et al., 2015), Laplace (Trzasko & Manduca, 2008), LNN (Peng et al., 2015), Logarithm (Friedman, 2012), ETP (Gao et al., 2011), and Geman (Geman & Yang, 1995). A summary of these relaxations and the corresponding penalty function $h$ can be found in (Hu et al., 2021).

### 2.2 OPTIMIZATION OF THE LOW-RANK REGULARIZATION

Matrix factorization and rank function optimization are two prominent approaches for optimizing models with low-rank structures.

The concept of matrix factorization involves decomposing the target matrix into the product of multiple low-rank components. Under this formulation, the optimization is straightforward. Notable examples include the decomposition of weight matrices into low-rank factors in works such as (Alvarez & Salzmann, 2017; Yang et al., 2020; Xu et al., 2021), which have found applications in neural network pruning and compression. Additionally, Geng et al. demonstrated that this strategy enables neural networks to learn global information and can even replace the attention mechanism (Geng et al., 2021). Ornhag et al. introduced a differentiable bilinear parameterization of the nuclear norm, relying on matrix decomposition. The authors also provided theoretical analyses regarding the optimality and convergence of the method (Örnhag et al., 2019). This idea was further extended in (Xu et al., 2017; Chen et al., 2021), where the multi-Schatten-$p$ norm was considered as a generalization of the bilinear parameterization. However, a significant challenge of this approach lies in the requirement of strong prior knowledge about the true rank of the matrix.

On the other hand, the rank function optimization approach introduces a regularization term to the loss function, allowing the appropriate matrix rank to be automatically determined during training. Initial research attempts applied existing convex optimization techniques to the problem, such as the Frank-Wolfe algorithm (Frank & Wolfe, 1956; Freund et al., 2017), proximal gradient algorithm (Ji & Ye, 2009; Yao et al., 2018), and the iteratively re-weighted algorithm (Mohan & Fazel, 2012; Kümmerle & Sigl, 2018; Nie et al., 2012). However, these methods require the loss function to be convex. Alternating direction method of multipliers (ADMM) (Boyd et al., 2011) is a popular method for optimizing the regularized loss, and has been successfully applied in various tasks (Gu et al., 2017; Li et al., 2015; Yang et al., 2016). Recently, it is discovered that one of the optimization sub-procedures within ADMM can be seen as performing image denoising, enabling the integration of existing denoising neural networks into the ADMM framework (Chan et al., 2016; Hu et al., 2022b; Liu et al., 2023). However, in these methods gradient-based optimization cannot be applied and high-performing GPUs cannot be fully utilized.

There are a few works that consider SVD-free optimization of LRRs, which draw inspirations from the variational characterization of the nuclear norm, $i.e.$, $\|\mathbf{X}\|_* = \min_{\mathbf{AB}=\mathbf{X}} \frac{1}{2}(\|\mathbf{A}\|_F^2 + \|\mathbf{B}\|_F^2)$, where $\mathbf{X} \in \mathbb{R}^{m\times n}$, $\mathbf{A} \in \mathbb{R}^{m\times d}$, $\mathbf{B} \in \mathbb{R}^{d\times n}$ and $d \geq rank(\mathbf{X})$ (Srebro et al., 2004; Rennie & Srebro, 2005). Recent research further extends these methods to handle the Schatten-$p$ quasi-norm, $i.e.$, $\|\mathbf{x}\|_p = (\sum_i |x_i|^p)^{\frac{1}{p}}$ with $p \in (0,1)$ (Shang et al., 2016; Fan et al., 2019; Giampouras et al., 2020). However, compared to our work, all these methods suffer the following critical drawbacks: 1) They explicitly require the upper bound of the true rank $d$ as input, which is a difficulty shared by the aforementioned factorization-based methods. When $d$ is too large, the slow convergence brings huge computational burden, while a too small $d$ deteriorates the performance. 2) These method can only optimize a restricted class of LRRs, particularly the nuclear norm and the Schatten-$p$ quasi-norm with $p \in (0,1)$. They cannot be applied to various recently proposed nonconvex LRRs like the $\gamma$-Nuclear norm (Kang et al., 2015) and Laplace (Trzasko & Manduca, 2008). In contrast, our proposed method is quite general, applicable to a much broader types of LRRs.

## 3 METHODOLOGY

### 3.1 NOTATIONS AND PROBLEM STATEMENT

**Notations** In this paper uppercase bold letters ($e.g.$, $\mathbf{X}$) denote matrices, and lowercase bold letters ($e.g.$, $\mathbf{x}$) denote vectors. For a vector $\mathbf{x}$, $\|\mathbf{x}\|_p = (\sum_i |x_i|^p)^{\frac{1}{p}}$ represents its $l_p$-norm. For a matrix $\mathbf{X}$, $\sigma_i(\mathbf{X})$ is its $i$th largest singular value, or abbreviated as $\sigma_i$. We use $\mathbf{X} \succeq 0$ to indicate matrix $\mathbf{X}$ is positive semi-definite. $\|\mathbf{X}\|_p = (\sum_{i=1}^{r} \sigma_i(\mathbf{X})^p)^{\frac{1}{p}}$ is the Schatten-$p$ norm, where $r$ is the rank of $\mathbf{X}$. Particularly, the nuclear norm is $\|\mathbf{X}\|_* = \|\mathbf{X}\|_1$, the spectral norm is $\|\mathbf{X}\| = \sigma_1(\mathbf{X})$, and the rank is equivalently represented as $\|\mathbf{X}\|_0$. $\mathbf{X}^\top$ and $\mathbf{X}^\dagger$ denote the transpose and the pseudo-inverse of the matrix respectively. $span(\mathbf{X})$ is the linear space spanned by the columns of $\mathbf{X}$, and $P_\mathbf{X}[\mathbf{u}] = \mathbf{X}\mathbf{X}^\dagger \mathbf{u}$ denotes the projection of the vector $\mathbf{u}$ onto the column space $span(\mathbf{X})$.

**Problem statement** For a given input-output pair $(\mathbf{X}, \mathbf{Y})$ where $\mathbf{X} \in \mathbb{R}^{a\times b}$ and $\mathbf{Y} \in \mathbb{R}^{c\times d}$, consider the empirical loss w.r.t. a learning function $f$ parameterized by $\boldsymbol{\theta}$, denoted as:

$$\mathcal{L}(\mathbf{X}, \mathbf{Y}, \boldsymbol{\theta}) = l(f(\mathbf{X}; \boldsymbol{\theta}), \mathbf{Y}) + \mathcal{R}(\mathbf{S}), \text{ where } \mathbf{S} = g(\mathbf{X}, \mathbf{Y}, \boldsymbol{\theta}) \in \mathbb{R}^{m\times n}. \quad (1)$$

Here the matrix $\mathbf{S}$ can depend on both the data and function parameters.

The regularization term $\mathcal{R}(\mathbf{S})$ enforces the matrix $\mathbf{S}$ to be low-rank. Ideally, the definition could be directly applied, *i.e.*, $\mathcal{R}(\mathbf{S}) = \|\mathbf{S}\|_0$. However, this regularization term is nondifferentiable, and the associated optimization problem is NP-hard in general (Wright & Ma, 2022). To address this problem, in this work $\mathcal{R}(\mathbf{S})$ will represent some form of approximation or relaxation of the rank. For example, it is well-known that the nuclear norm (*i.e.*, $\|\mathbf{S}\|_* = \sum_{i=1}^{r} \sigma_i(\mathbf{S})$) is the convex envelope of the rank function (Wright & Ma, 2022), and is used as the surrogate function in many works. However, the nuclear norm penalizes all singular values simultaneously, while in practical problems it is desired that large singular values are less impacted, so the important information of the matrix is preserved. Motivated by this observation, many nonconvex relaxations of the rank function have been introduced (Kang et al., 2015; Peng et al., 2015; Friedman, 2012; Gao et al., 2011). To encompass all these formulations, in this work the regularization term is considered to be of the form:

$$\mathcal{R}(\mathbf{S}) = \sum_{i=1}^{r} h(\sigma_i(\mathbf{S})). \tag{2}$$

Here function $h$ generally increases the penalty of small singular values. In this paper, we refer to this form of regularization as the **generalized low-rank regularization (LRR)**.

The goal of this paper is to develop a method to approximately compute the generalized LRR $\mathcal{R}(\mathbf{S})$ in Eq. (2), which is a differentiable operation that allows the gradients to be computed and propagated. Thus $\mathcal{R}(\mathbf{S})$ can be conveniently used in a plug-and-play fashion, and the optimization problem in Eq. (1) can be conveniently solved by off-the-shelf optimization frameworks, and utilize high-performing GPUs for efficient parallel computation.

To better illustrate the problem definition, several concrete examples are presented below:

**Matrix completion**     Suppose $\mathbf{S} \in \mathbb{R}^{m \times n}$ is a low-rank matrix, $\Omega \subset \{1, .., m\} \times \{1, ..., n\}$ is an index set that denotes the observable entries, and $P_\Omega[\mathbf{S}]_{ij} = \mathbf{1}_{\{(i,j) \in \Omega\}} \cdot \mathbf{S}_{ij}$ represents the observation projection. The problem of matrix completion aims to recover $\mathbf{S}$ based on partial observations, *i.e.*, $\min_{\mathbf{X}} \|P_\Omega[\mathbf{X}] - P_\Omega[\mathbf{S}]\|_F^2 + \lambda \mathcal{R}(\mathbf{X})$.

**Video fore-background separation**     A video sequence is represented as a 3D tensor $\mathbf{V} \in \mathbb{R}^{a \times b \times t}$, where $a$, $b$ denote the width and height of each frame, and $t$ indexes time. Let $\mathbf{V}' \in \mathbb{R}^{ab \times t}$ be a reshaped 2D matrix. In video fore-background separation it is assumed that the reshaped matrix can be decomposed as $\mathbf{V}' = \mathbf{S} + \mathbf{O}$, where $\mathbf{S}$ is a low-rank matrix that represents the background, and $\mathbf{O}$ is a sparse matrix that represents the foreground object. So the problem can be solved by optimizing $\min_{\mathbf{X}} \|\mathbf{V}' - \mathbf{X}\|_1 + \lambda \mathcal{R}(\mathbf{X})$.

**DNN-based image denoising**     In image denoising an observed image $\mathbf{X}$ is considered as a clean image $\mathbf{S}$ corrupted by noise $\mathbf{N}$, *i.e.*, $\mathbf{X} = \mathbf{S} + \mathbf{N}$. A deep neural network (DNN) based model trains a function $f$ to predict the noise by optimizing the loss $\min_{\boldsymbol{\theta}} E \left[ \|f(\mathbf{X}; \boldsymbol{\theta}) - \mathbf{N}\|_F^2 \right]$. It is known that the clean images are approximately low-rank, so it is desirable to utilize this prior information during training, and regularize the loss function to be $\min_{\boldsymbol{\theta}} E \left[ \|f(\mathbf{X}; \boldsymbol{\theta}) - \mathbf{N}\|_F^2 + \lambda \mathcal{R}(f(\mathbf{X}; \boldsymbol{\theta}) - \mathbf{N}) \right]$. Note that most existing work has difficulty in optimizing such general formulations.

### 3.2 Differentiable and parallelizable Low-rank Regularization

#### 3.2.1 Iterative methods for matrix pseudo-inverse and square root

We first show that two fundamental operations, *i.e.*, matrix pseudo-inverse and matrix square root, have differentiable approximations. These results serve as the cornerstones of the proposed method, as we later demonstrate that the general LRR can be constructed with these operations.

**Proposition 1** (Iterative method for matrix pseudo-inverse (Ben-Israel & Cohen, 1966)). *For a given matrix* $\mathbf{S} \in \mathbb{R}^{m \times n}$, *define the recursive sequence* $\mathbf{S}_{i+1} = 2\mathbf{S}_i - \mathbf{S}_i \mathbf{S} \mathbf{S}_i$, *with* $\mathbf{S}_0 = \alpha \mathbf{S}^\top$. *Then* $\lim_{i \to \infty} \mathbf{S}_i = \mathbf{S}^\dagger$, *provided* $0 < \alpha < 2/\sigma_1^2(\mathbf{S})$.

Intuitively, when the iteration converges we have $\mathbf{S}_i = \mathbf{S}_{i+1}$. So $\mathbf{S}_i = \mathbf{S}_i \mathbf{S} \mathbf{S}_i$, which is exactly the definition of pseudo-inverse.

In practice, taking a large enough iteration step $N$, results in a satisfactory approximation of the matrix's pseudo inverse, *i.e.*, $\mathbf{S}_N \approx \mathbf{S}^\dagger$. Furthermore, since the iterative computation only involves matrix multiplication and subtraction, it is obviously a differentiable operation.

As a consequence, the projection operator $P_{\mathbf{S}}[\mathbf{u}]$ also has a differentiable approximation. Recall that $P_{\mathbf{S}}[\mathbf{u}] = \mathbf{SS}^{\dagger}\mathbf{u}$ denotes the projection of the vector $\mathbf{u}$ onto the column space of matrix $\mathbf{S}$. So $P_{\mathbf{S}}[\mathbf{u}] \approx \mathbf{SS}_N\mathbf{u}$, where $\mathbf{S}_N$ is the approximation of the pseudo-inverse computed by the iterative method. Again the r.h.s. is obviously differentiable.

The matrix square root can also be computed by an iterative method, which is called the Newton-Schulz iteration (NS iteration) (Schulz, 1933; Song et al., 2022):

**Proposition 2** (NS iteration for matrix square root). *For $\mathbf{A} \in \mathbb{R}^{m \times m}$, initialize $\boldsymbol{Y}_0 = \frac{1}{\|\boldsymbol{A}\|_{\mathrm{F}}}\boldsymbol{A}, \boldsymbol{Z}_0 = \mathbf{I}$. The Newton-Schulz method defines the following coupled iteration:*

$$\boldsymbol{Y}_{k+1} = \frac{1}{2}\boldsymbol{Y}_k\left(3\boldsymbol{I} - \boldsymbol{Z}_k\boldsymbol{Y}_k\right), \boldsymbol{Z}_{k+1} = \frac{1}{2}\left(3\boldsymbol{I} - \boldsymbol{Z}_k\boldsymbol{Y}_k\right)\boldsymbol{Z}_k.$$

*Then $\sqrt{\|\boldsymbol{A}\|_{\mathrm{F}}}\boldsymbol{Y}_k$ quadratically converges to $\boldsymbol{A}^{\frac{1}{2}}$, i.e., $\sqrt{\|\boldsymbol{A}\|_{\mathrm{F}}}\boldsymbol{Y}_k \to \boldsymbol{A}^{\frac{1}{2}}$.*

We highlight the following points of Proposition 1 and 2, which present iterative methods for computing the matrix pseudo inverse and matrix square root: 1) Both are **differentiable** operations, which allow gradient-based methods for the associated optimization; 2) Both are **parallelizable** operations, which are GPU-friendly and thus efficient for large-scale datasets. Our proposed method inherits these advantages, which results in differentiable and parallelizable approximation of the generalized LRR.

### 3.2.2 DIFFERENTIABLE RANK APPROXIMATION

The rank of a matrix is defined as the dimension of its column space (or equivalently the row space). The most well-known method for computing the rank is to first apply SVD, and then count the number of nonzero singular values. However, the SVD step is nondifferentiable. Interestingly, the following proposition presents an alternative approach for computing ranks without using SVD. Due to space limitation, all the proofs are deferred to the Appendix.

**Proposition 3** (Equivalent definition of matrix rank (Wright & Ma, 2022)). *The rank of a matrix $\mathbf{S}$ can be equivalently computed as the average squared length of a random Gaussian vector (i.e., $\mathbf{g} \sim \mathcal{N}(\mathbf{0}, \mathbf{I})$) projected onto the column space of $\mathbf{S}$:*

$$\|\mathbf{S}\|_0 = rank(\mathbf{S}) = E\left[\|P_{\mathbf{S}}[\mathbf{g}]\|_2^2\right]. \tag{3}$$

To apply the result for practical rank computation, first sample $N$ independent random Gaussian vectors $\mathbf{g}_1, ..., \mathbf{g}_N \sim \mathcal{N}(\mathbf{0}, \mathbf{I})$. Then the sample average is used to approximate the rank, *i.e.*, $\|\mathbf{S}\|_0 = E\left[\|P_{\mathbf{S}}[\mathbf{g}]\|_2^2\right] \approx \frac{1}{N}\sum_{i=1}^{N}\|P_{\mathbf{S}}[\mathbf{g}_i]\|_2^2$. Since it has been shown that the projection operator $P_{\mathbf{S}}[\mathbf{g}]$ has a differentiable approximation, by substituting the routine we obtain a differentiable method for calculating the matrix rank.

However, rank is a piecewise constant function. Though it is now differentiable, it cannot provide useful gradient information for the overall optimization problem. To address this challenge, in the following we consider several relaxations of matrix rank, the approximations of which are not only differentiable, but also provide informatic gradients.

### 3.2.3 DIFFERENTIABLE NUCLEAR NORM APPROXIMATION

The nuclear norm is the convex envelope of rank, and is the most popular relaxation. The common approach for computing the nuclear norm is to first apply SVD, and then sum up the singular values. The following proposition presents an alternative method, similar to the case of rank calculation.

**Proposition 4.** *The nuclear norm of a matrix $\mathbf{S}$ can be equivalently computed as:*

$$\|\mathbf{S}\|_* = E\left[\langle P_{\mathbf{S}}[\mathbf{g}], (\mathbf{SS}^{\top})^{\frac{1}{2}}\mathbf{g}\rangle\right], \tag{4}$$

*where $\mathbf{g} \sim \mathcal{N}(\mathbf{0}, \mathbf{I})$ is a random Gaussian vector.*

Note that besides the projection operator $P_{\mathbf{S}}[\mathbf{g}]$, this proposition further requires calculating the matrix square root $(\mathbf{SS}^{\top})^{\frac{1}{2}}$, which can be obtained with Proposition 2. Finally, by replacing the expectation with the sample mean, the nuclear norm can be approximately computed using differentiable operators. Furthermore, the approximated nuclear norm can provide useful gradient information for optimization.

### 3.2.4 DIFFERENTIABLE APPROXIMATION OF THE GENERALIZED LOW-RANK REGULARIZATION

In this subsection, we consider general LRR of the form $\mathcal{R}(\mathbf{S}) = \sum_{i=1}^{r} h(\sigma_i(\mathbf{S}))$, where $r$ denotes the rank of $\mathbf{S}$. The function $h$ increases the penalty of the small singular values, so that the principle information of the matrix is preserved while reducing the rank.

First, we introduce a lemma that allows any Schatten-$p$ norm to be stochastically computed. This lemma can be viewed as an extension of Proposition 3 and 4, and the proof is also similar.

**Lemma 1.** *For a matrix $\mathbf{S}$, its Schatten-$p$ norm, defined as $\|\mathbf{S}\|_p = (\sum_{i=1}^{r} \sigma_i(\mathbf{S})^p)^{\frac{1}{p}}$ where $r$ is the rank of $\mathbf{S}$, can be alternatively computed as*

$$\|\mathbf{S}\|_p^p = \sum_{i=1}^{r} \sigma_i^p = E\left[\langle P_{\mathbf{S}}\left[\mathbf{g}\right], (\mathbf{S}\mathbf{S}^\top)^{\frac{p}{2}}\mathbf{g}\rangle\right], \tag{5}$$

*where $p \in \mathbb{N}^+$ and $\mathbf{g} \sim \mathcal{N}(\mathbf{0}, \mathbf{I})$ is a random Gaussian vector.*

We are now ready to present the main result of this paper, which implies that a broad range of relaxations of the rank function can be computed with a stochastic method. Particularly, we introduce two methods to utilize the above lemma, which are based on the Taylor expansion and the Laguerre expansion respectively.

**Taylor Expansion-based Generalized Low-rank Regularization**

**Theorem 1.** *Let $\mathbf{S}$ be a matrix of rank $r$, and $h : \mathbb{R} \to \mathbb{R}$ be a sufficiently smooth function and $\mathbf{g} \sim \mathcal{N}(\mathbf{0}, \mathbf{I})$. Then the generalized LRR defined in Eq. (2) can be computed as*

$$\sum_{i=1}^{r} h(\sigma_i(\mathbf{S})) = \sum_{p=0}^{\infty} \frac{h^{(p)}(0)}{p!} E\left[\langle P_{\mathbf{S}}\left[\mathbf{g}\right], (\mathbf{S}\mathbf{S}^\top)^{\frac{p}{2}}\mathbf{g}\rangle\right]. \tag{6}$$

However, Taylor expansion approximates the target function based on a fixed initial point, and the truncated error grows when the evaluation location moves away from the initial point. This motivates the application of advanced approximation techniques.

**Laguerre Expansion-based Generalized Low-rank Regularization**

Orthogonal polynomial approximation is an important family of function approximation techniques, including the well-known Hermite expansion, Legendre expansion and Laguerre expansion (Szeg, 1939; Koornwinder et al., 2010). In our problem the Laguerre expansion is particularly suitable, since it approximates functions with range $(0, +\infty)$.

The Laguerre expansion is based on the Laguerre polynomials $\mathfrak{L} = \{L_1(x), L_2(x), ...\}$, which is a countable infinite set of mutually orthogonal polynomials, each .denoted as $L_k(x) = \sum_p a_{k,p} x^p$. For a target function $f(x)$ with range $(0, +\infty)$, the method decomposes it into $f(x) = \sum_{k \geq 0} c_k L_k(x)$, where $c_k$ are the coefficients computed as $c_k = \int_0^\infty L_k(x) e^{-x} f(x) \mathrm{d}x$.

**Theorem 2.** *Keeping the same notations, we have*

$$\sum_{i=1}^{r} h(\sigma_i(\mathbf{S})) = \sum_{k \geq 0} \sum_p c_k a_{k,p} E\left[\langle P_{\mathbf{S}}\left[\mathbf{g}\right], (\mathbf{S}\mathbf{S}^\top)^{\frac{p}{2}}\mathbf{g}\rangle\right]. \tag{7}$$

*Here $h(x) = \sum_{k \geq 0} c_k L_k(x)$ utilizes the Laguerre expansion, $a_{k,p}$ are the polynomial coefficients.*

In practice, the required coefficients can be computed by off-the-shelf tools like Mathematica.

In both Theorem 1 and 2, by using a finite sum to approximate the infinite series, the general regularization form becomes differentiable, and can provide useful gradients for the optimization. Furthermore, the computation solely depends on matrix multiplication, which is a GPU-friendly operation that allows highly efficient parallel implementation.

## 4 EXPERIMENTAL RESULTS

In this section, we perform various experiments to demonstrate the versatility, convenience, as well as efficiency of our method. We first examine two classic LRR tasks, *i.e.*, matrix completion and video

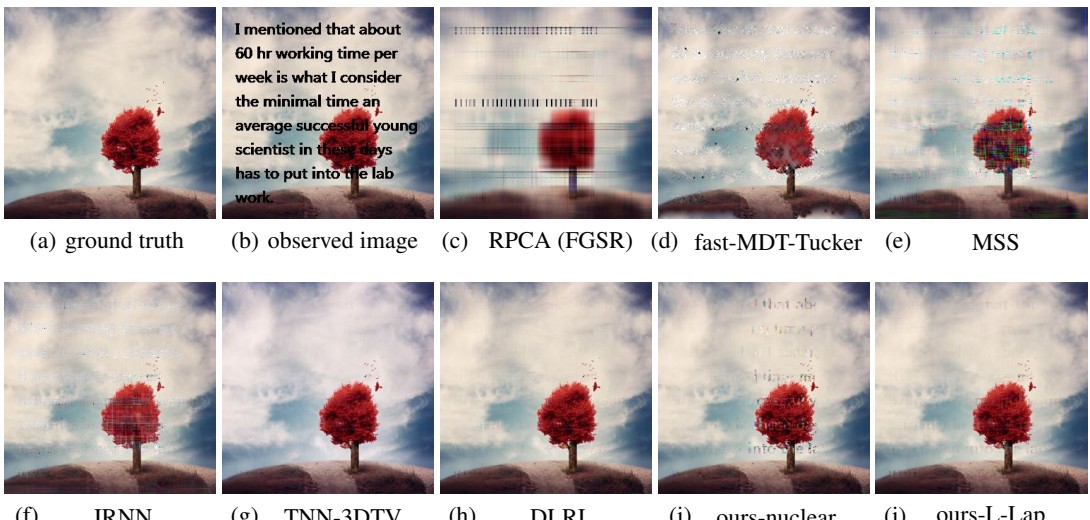

Figure 1: Comparison of matrix completion algorithms for the text removal problem. (a) ground truth (b) image with text (c)-(j) recovered images.

fore-background separation. One advantage of our proposed method is to conveniently introduce LRR terms into any loss function, particularly deep neural networks. So we further exploit this property in DNN-based image denoising. A detailed convergence and parameter sensitivity analysis is deferred to Appendix A.2.1 due to space constraints. All experiments were conducted on a machine equipped with 3080Ti GPU.

## 4.1 MATRIX COMPLETION

Table 1: Comparison of matrix completion algorithms for image inpainting. Bold **terms** and underlined terms denote the best and second best results. "time" denotes the average processing time per image. See the main text for details.

| Method
PSNR | nuclear | T-$\gamma^*$ | T-Lap | L-$\gamma^*$ | L-Lap | RPCA
FGSR | f-MDT
Tucker | MSS | IRNN | TNN-
3DTV | DLRL |
|---|---|---|---|---|---|---|---|---|---|---|---|
| drop 20% | 36.85 | 38.57 | **38.58** | 38.37 | 38.47 | 26.0 | 24.28 | 26.62 | 28.33 | 30.17 | 35.99 |
| drop 30% | 35.13 | 35.92 | 36.16 | **36.38** | 36.23 | 24.89 | 24.44 | 25.58 | 28.51 | 30.03 | 34.58 |
| drop 40% | 33.28 | 34.13 | **34.36** | 34.35 | 34.28 | 16.41 | 24.16 | 24.83 | 27.52 | 29.82 | 33.58 |
| drop 50% | 31.46 | **32.48** | 32.25 | 32.39 | 32.39 | 6.95 | 23.90 | 24.05 | 26.91 | 29.54 | 32.43 |
| block | 17.89 | 30.55 | 29.67 | 24.68 | 25.80 | 13.75 | 23.20 | 26.09 | 25.46 | 28.17 | **30.82** |
| text | 29.51 | 34.90 | 34.98 | 34.99 | 35.01 | 21.49 | 22.68 | 24.56 | 26.20 | 29.99 | **37.19** |
| time (s) | 4.35 | 3.83 | 3.88 | 3.83 | 3.86 | 16.83 | 1.73 | 10.70 | 8.82 | 24.85 | 494.46 |

Natural images can be represented as matrices that possess low-rank priors. Particularly, singular values of natural images are dominated by a few of the largest components, which allows us to model image restoration as a low-rank matrix completion problem. When dealing with colorful images, we process each color channel as an independent matrix. We evaluate the effectiveness of different methods using the PSNR.

Table 1 presents the numerical result, and Figure 1 visualizes the image qualities. In Table 1, the first five columns denote our method, using different approximation and relaxation strategies. Particularly, "nuclear" denotes using the nuclear norm without other relaxation function, *i.e.*, directly apply Proposition 4. "T-$\gamma^*$" and "T-Lap" denote using the Taylor expansion-based method for function approximation, with the $\gamma$-nuclear norm (Kang et al., 2015) and Laplace (Trzasko & Manduca, 2008; Hu et al., 2021) as the relaxation function (*i.e.*, $h$) respectively. Similarly, "L-$\gamma^*$" and "L-Lap" denote using the Laguerre expansion-based method for function approximation. "drop XX%" means randomly removing a certain potion of pixels, while "block" and "text" use predefined patterns

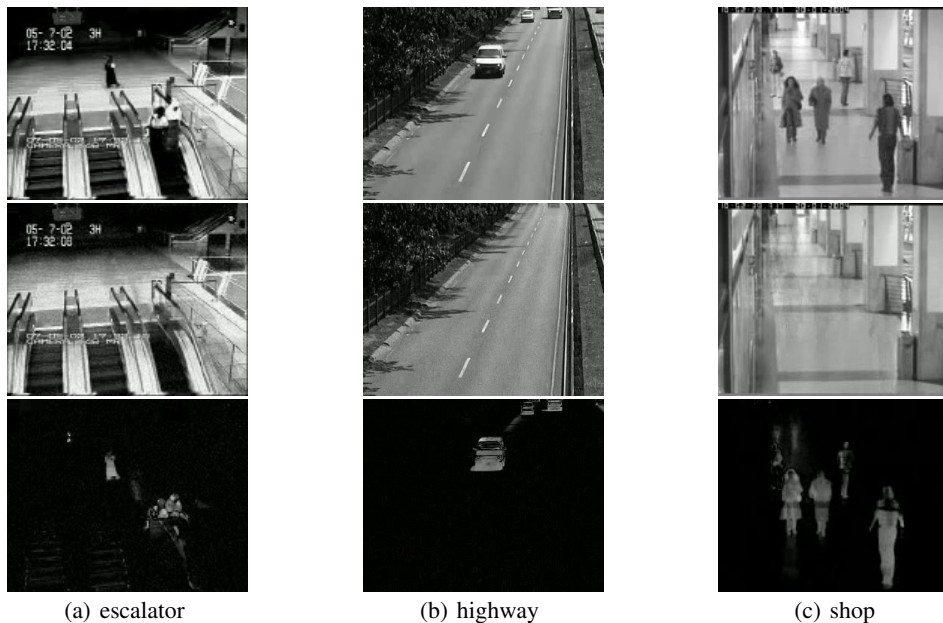

(a) escalator          (b) highway          (c) shop

Figure 2: The results using our algorithm in fore-background separation. From top to bottom: original frames of the video, separated backgrounds, and foreground objects.

to obscure the image. The methods being compared include the fast Multiway Delay-embedding Transform (MDT) tucker method (Yamamoto et al., 2022), Multi-Schatten-$p$ norm Surrogate (MSS) (Oh et al., 2015), Iteratively Reweighted Nuclear Norm for Nonconvex Nonsmooth Low-rank Minimization (IRNN) (Lu et al., 2015), Robust Principal Components Analysis (RPCA) Factor Group-Sparse Regularization (FGSR) (Fan et al., 2019), Differentiable Low-Rank Learning (DLRL) (Chen et al., 2021), and Tensor Nuclear Norm (TNN-3DTV) (Jiang et al., 2018).

From the results we can see that: 1) Our method outperforms other baselines in almost all the cases; 2) By using relaxation methods, our performance can be further improved; 3) Both Taylor and Laguerre expansions are effective for function approximation, with each excelled in different scenarios; 4) By utilizing parallel computation and high-performing GPUs, our method strikes the best balance between performance and efficiency.

Further results and visualization of matrix completion can be found in the Appendix.

## 4.2 VIDEO FORE-BACKGROUND SEPARATION

We now apply our method to the task of video fore-background separation. Given a video sequence $\mathbf{V} \in \mathbb{R}^{a \times b \times t}$, where $a$ and $b$ represent the dimensions of each frame, and $t$ indexes the time steps. For each frame $\mathbf{f} \in \mathbb{R}^{a \times b}$, we can reshape the matrix into a vector $\mathbf{f}' \in \mathbb{R}^{ab \times 1}$ and concatenate all frames together, resulting in the final matrix $\mathbf{V}' \in \mathbb{R}^{ab \times t}$. We assume that the reshaped matrix can be decomposed as $\mathbf{V}' = \mathbf{S} + \mathbf{O}$, where $\mathbf{S}$ is a low-rank matrix representing the background, and $\mathbf{O}$ is a sparse matrix representing the foreground object. Thus, the problem can be solved by optimizing $\min_{\mathbf{X}} \|\mathbf{V}' - \mathbf{X}\|_1 + \lambda \mathcal{R}(\mathbf{X})$, where $\| \cdot \|_1$ denotes the L1 norm and $\mathcal{R}(\mathbf{X})$ represents the LRR term. We use the nuclear norm as the surrogate rank function.

Figure 2 illustrates the results obtained by applying our method to fore-background separation. The algorithm effectively separates the backgrounds of individual frames by utilizing the global information of the complete video sequences. Notably, our approach produces distinct boundaries for the background, even in dynamic scenarios such as the continuously moving escalator example. Unlike conventional methods that tend to blur the details of moving objects like escalator steps, our algorithm maintains clear edges and boundaries in the obscured background region.

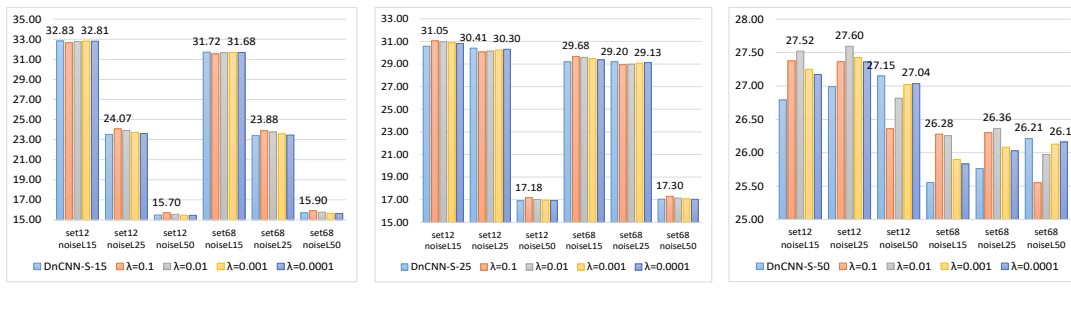

(a)  trained with noise level 15   (b)  trained with noise level 25   (c)  trained with noise level 50

Figure 3: Results of applying low-rank regularization and the proposed differentiable approximation technique in the DnCNN denoising model, measured in PSNR.

### 4.3 REGULARIZING DNN-BASED DENOISING MODELS

One of the major strengths of our proposed method is its flexibility in incorporating the LRR term into any loss function, and the optimization can be accomplished with machine learning libraries. Given the impressive performance of deep neural networks, it is highly desirable to apply our approach to leverage low-rank priors. In this experiment, we explore this avenue in DNN-based image denoising, particularly using the denoising convolutional neural networks (DnCNNs) (Zhang et al., 2017).

Following the experimental configuration of (Zhang et al., 2017), we conducted our experiments using a training set of 400 images with dimensions of $180 \times 180$. Gaussian noise levels were set at $\sigma = 15, 25, 50$. Since the low-rank structure is only applicable to the entire image, we utilized the full image as input when calculating the regularization loss. For the reconstruction loss, the settings remained unchanged, with the images divided into patches of size $40 \times 40$. The datasets in our experiments were the Berkeley segmentation dataset (BSD68) and the Set12 dataset, which were consistent with previous studies.

The comparison between the original network and the network augmented with LRR is depicted in Figure 3. One of the challenges faced by these denoising networks is their reliance on prior knowledge of the noise level ($\sigma$) during training. Consequently, the trained models tend to overfit to the specific noise level provided, resulting in inferior performance when confronted with different noise levels during testing. However, as observed in the figure, this issue is significantly mitigated when the LRR and our proposed differentiable approximation are applied. Although there is a slight drop in performance when the test noise level matches the training noise level, substantial gains are observed at unseen noise levels. This is most evident in Figure 3 (c), in which our method frequently improve the performance by a large margin.

## 5 CONCLUSION

In this paper, a novel differentiable approximation of the generalized LRR was proposed, which is based on the key observation that the rank function and its relaxations have equivalent stochastic definitions. The form of the regularization considered in our work is quite general, covering a broad range of both convex and nonconvex relaxations. The key advantages of the proposed method include its versatility, convenience, and efficiency. By appending the differentiable LRR to a loss function in a plug-and-play fashion, the optimization can be automatically accomplished with gradient-based machine learning libraries. Moreover, our method solely relies on matrix multiplication, an operation well-suited for GPU acceleration and facilitates efficient parallel implementation. In the experimental study, the proposed method was successfully applied to a variety of tasks.

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

# A APPENDIX

## A.1 PROOFS

**Proposition 3** (Equivalent definition of matrix rank (Wright & Ma, 2022)). *The rank of a matrix* $\mathbf{S}$ *can be equivalently computed as the average squared length of a random Gaussian vector (i.e.,* $\mathbf{g} \sim \mathcal{N}(\mathbf{0}, \mathbf{I})$*) projected onto the column space of* $\mathbf{S}$*:*

$$\|\mathbf{S}\|_0 = rank(\mathbf{S}) = E\left[\|P_\mathbf{S}[\mathbf{g}]\|_2^2\right]. \tag{8}$$

*Proof.* Suppose the rank of matrix $\mathbf{S} \in \mathbb{R}^{m \times n}$ is $rank(\mathbf{S}) = r$. By applying (compact) SVD, $\mathbf{S} = \mathbf{U}\boldsymbol{\Sigma}\mathbf{V}^\top$, with $\mathbf{U} \in \mathbb{R}^{m \times r}, \boldsymbol{\Sigma} \in \mathbb{R}^{r \times r}, \mathbf{V} \in \mathbb{R}^{n \times r}$. Also recall that the pseudo-inverse of matrix $\mathbf{S}$ satisfies $\mathbf{S}^\dagger = \mathbf{V}\boldsymbol{\Sigma}^{-1}\mathbf{U}^\top$. Thus,

$$E\left[\|P_\mathbf{S}[\mathbf{g}]\|_2^2\right] = E\left[(\mathbf{S}\mathbf{S}^\dagger\mathbf{g})^\top(\mathbf{S}\mathbf{S}^\dagger\mathbf{g})\right] = E\left[\|\mathbf{U}^\top\mathbf{g}\|_2^2\right] = E\left[(\mathbf{u}_1^\top\mathbf{g})^2\right] + ... + E\left[(\mathbf{u}_r^\top\mathbf{g})^2\right].$$

For any $\mathbf{u} \in \{\mathbf{u}_1, ..., \mathbf{u}_r\}$, $E\left[(\mathbf{u}^\top\mathbf{g})^2\right] = \sum_{i,j=1}^m u_i u_j E\left[g_i g_j\right] = \sum_{i=1}^m u_i^2 E\left[g_i^2\right] = 1$. So $\|\mathbf{S}\|_0 = r = E\left[\|P_\mathbf{S}[\mathbf{g}]\|_2^2\right]$. □

**Proposition 4.** *The nuclear norm of a matrix* $\mathbf{S}$ *can be equivalently computed as:*

$$\|\mathbf{S}\|_* = E\left[\langle P_\mathbf{S}[\mathbf{g}], (\mathbf{S}\mathbf{S}^\top)^{\frac{1}{2}}\mathbf{g}\rangle\right], \tag{9}$$

*where* $\mathbf{g} \sim \mathcal{N}(\mathbf{0}, \mathbf{I})$ *is a random Gaussian vector.*

*Proof.* The proof is similar to that of Proposition 3. Keeping the same notations, it follows that

$$E\left[\langle P_\mathbf{S}[\mathbf{g}], (\mathbf{S}\mathbf{S}^\top)^{\frac{1}{2}}\mathbf{g}\rangle\right] = E\left[\mathbf{g}^\top\mathbf{S}\mathbf{S}^\dagger(\mathbf{S}\mathbf{S}^\top)^{\frac{1}{2}}\mathbf{g}\right] = E\left[\|\boldsymbol{\Sigma}^{\frac{1}{2}}\mathbf{U}^\top\mathbf{g}\|_2^2\right]$$

$$= \sigma_1 E\left[(\mathbf{u}_1^\top\mathbf{g})^2\right] + ... + \sigma_r E\left[(\mathbf{u}_d^\top\mathbf{g})^2\right] = \sum_{i=1}^r \sigma_i = \|\mathbf{S}\|_*. \quad \square$$

**Theorem 1.** *Let* $\mathbf{S}$ *be a matrix of rank* $r$, *and* $h : \mathbb{R} \to \mathbb{R}$ *be a sufficiently smooth function and* $\mathbf{g} \sim \mathcal{N}(\mathbf{0}, \mathbf{I})$. *Then the generalized LRR defined in Eq. (2) can be computed as*

$$\sum_{i=1}^r h(\sigma_i(\mathbf{S})) = \sum_{p=0}^\infty \frac{h^{(p)}(0)}{p!} E\left[\langle P_\mathbf{S}[\mathbf{g}], (\mathbf{S}\mathbf{S}^\top)^{\frac{p}{2}}\mathbf{g}\rangle\right]. \tag{10}$$

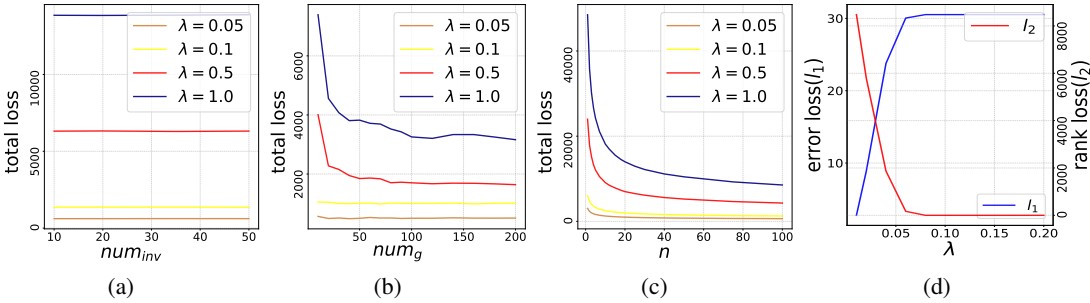

Figure 4: Numerical analysis on the synthetic dataset.

*Proof.* The result can be easily proved by using Lemma 1 and Taylor expansion.

$$\sum_{i=1}^{r} h(\sigma_i(\mathbf{S})) = \sum_{i=1}^{r}\sum_{p=0}^{\infty} \frac{h^{(p)}(0)}{p!}\sigma_i^p = \sum_{p=0}^{\infty} \frac{h^{(p)}(0)}{p!} E\left[\langle P_{\mathbf{S}}\left[\mathbf{g}\right], (\mathbf{S}\mathbf{S}^{\top})^{\frac{p}{2}}\mathbf{g}\rangle\right]. \qquad \square$$

**Theorem 2.** *Keeping the same notations, we have*

$$\sum_{i=1}^{r} h(\sigma_i(\mathbf{S})) = \sum_{k\geq 0}\sum_{p} c_k a_{k,p} E\left[\langle P_{\mathbf{S}}\left[\mathbf{g}\right], (\mathbf{S}\mathbf{S}^{\top})^{\frac{p}{2}}\mathbf{g}\rangle\right]. \tag{11}$$

*Here $h(x) = \sum_{k\geq 0} c_k L_k(x)$ utilizes the Laguerre expansion, $a_{k,p}$ are the polynomial coefficients.*

*Proof.* The result can be obtained by:

$$\sum_{i=1}^{r} h(\sigma_i(\mathbf{S})) = \sum_{i=1}^{r}\sum_{k\geq 0} c_k L_k(\sigma_i) = \sum_{i=1}^{r}\sum_{k\geq 0}\sum_{p} c_k a_{k,p}\sigma_i^p =$$
$$= \sum_{k\geq 0}\sum_{p} c_k a_{k,p} E\left[\langle P_{\mathbf{S}}\left[\mathbf{g}\right], (\mathbf{S}\mathbf{S}^{\top})^{\frac{p}{2}}\mathbf{g}\rangle\right]. \qquad \square$$

## A.2 MORE EXPERIMENTAL RESULTS

### A.2.1 CONVERGENCE AND PARAMETER SENSITIVITY ANALYSIS

We perform experiments on a synthetic dataset, the goal of which is to analyze the convergence and parameter sensitivity of the proposed method. The synthetic data is construed by sampling $\mathbf{A} \in \mathbb{R}^{m\times r}, \mathbf{B} \in \mathbb{R}^{r\times n}$, and $\mathbf{E} \in \mathbb{R}^{m\times n}$, the entries of each matrix follow i.i.d. Gaussian distribution ($m = n = 30, r = 30$). We define $\mathbf{C} = \mathbf{AB}, \mathbf{S} = \mathbf{C} + \sigma\mathbf{E}$, where $\mathbf{C}$ is the underlying low-rank matrix need to be discovered, and $\mathbf{S}$ represents the observed full-rank matrix obstructed by the noise $\sigma\mathbf{E}$. With this formulation, the target low-rank matrix can be found by solving:

$$l = \min_{\mathbf{X}} \underbrace{\|\mathbf{S} - \mathbf{X}\|_F^2}_{l_1:\text{ reconstruction loss}} + \underbrace{\lambda\mathcal{R}(\mathbf{X})}_{l_2:\text{ low-rank regularization}}. \tag{12}$$

Here we decompose the total loss $l$ into the reconstruction loss $l_1$ and the regularization loss $l_2$, and $\lambda$ controls the strength of regularization. In this experiment we consider the regularization to be the nuclear norm, *i.e.*, $\mathcal{R}(\mathbf{X}) = \|\mathbf{X}\|_*$. In our method, the following parameters play significant roles:

$num_{inv}$: In Proposition 1, the pseudo-inverse of a matrix is approximated by finite iterations. The iteration step is denoted as $num_{inv}$.
$num_g$: In Proposition 2, the matrix square root is approximated by a finite power series. The number of the summation terms is denoted as $num_g$.
$n$: In Proposition 4, the expectation will be approximated by an average of finite samples. The number of samples is denoted as $n$.

The sensitivity analysis results of these key parameters are shown in Figure 4 (a)-(c), from which we can see that: 1) Surprisingly, the choice of $num_{inv}$ has little impact on the convergence, and 10 iterations are sufficient for approximating the pseudo-inverse. A probable explanation is that gradient-based optimization involves many iteration steps, so its requirement for per-step accuracy is not strict. 2) The convergence is quite stable w.r.t. $num_g$ and $n$, as long as their values are reasonably large ($num_g \geq 100, n \geq 80$). We further vary parameter $\lambda$ to control the strength of regularization. As shown in Figure 4 (d), this parameter can effectively control the tradeoff between the reconstruction loss and the regularization loss as expected.

### A.2.2 MORE RESULTS FOR MATRIX COMPLETION

Table 2: More results for comparison of matrix completion algorithms for image inpainting.

| Method
PSNR | RPCA
nuclear norm | RPCA
F-nuclear norm | SNN | TNN |
|---|---|---|---|---|
| drop 20% | 27.50 | 27.49 | 30.14 | 30.09 |
| drop 30% | 23.70 | 23.70 | 29.95 | 29.87 |
| drop 40% | 12.09 | 12.08 | 29.68 | 29.56 |
| drop 50% | 6.40 | 6.40 | 29.23 | 29.03 |
| block | 13.73 | 13.73 | 27.54 | 27.84 |
| text | 22.25 | 22.26 | 29.81 | 29.81 |
| time (s) | 27.64 | 5.16 | 6.73 | 9.08 |

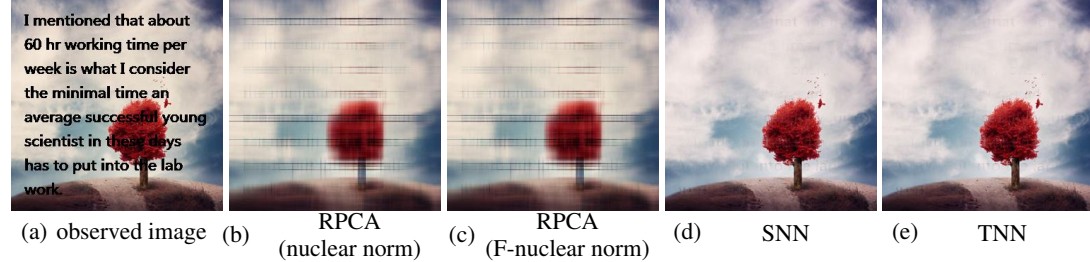

(a) observed image (b) RPCA (nuclear norm) (c) RPCA (F-nuclear norm) (d) SNN (e) TNN

Figure 5: More text inpaiting results. (a) image with text (b)-(e) recovered images.

Table 2 and Figure 5, following the same pattern of Table 1 and Figure 1, present more results of matrix completion algorithms for image inpaiting task. Specifically, RPCA (nuclear norm) and RPCA (F-nuclear norm) are weaker versions of RPCA (FGSR) (Fan et al., 2019). Sum of the Nuclear Norm (SNN) (Liu et al., 2012) and Tensor Nuclear Norm (TNN) (Zhang et al., 2014) are weaker versions of TNN-3DTV.

To further investigate the performance of our method utilizing nuclear norm and $h$ functions, we present the experimental results in Figure 6, 7, and 8. These figures showcase a subset of the conducted experiments where we applied both algorithms to images with various masking scenarios, enabling us to evaluate their performance across different conditions. The $h$ function we used in this experiment is Laguerre expansion-based Laplace function. Our findings consistently demonstrate that our method incorporating $h$ functions yields superior results in all image inpainting tasks. This substantiates the efficacy of $h$ functions, which approximate the pure rank, when employed in our approach.

Remarkably, even when confronted with challenging scenarios such as substantial and contiguous image regions being blocked, our method adeptly fills the missing parts based solely on the low-rank prior, thus providing plausible solutions. Notably, Figure 7 illustrates the block recovery problem, where our method with nuclear norm failed to achieve the desired peak signal-to-noise ratio (PSNR). However, it still exhibited a relatively high structural similarity index (SSIM), indicating that the algorithm successfully generated missing parts that closely resemble the original image's structure.

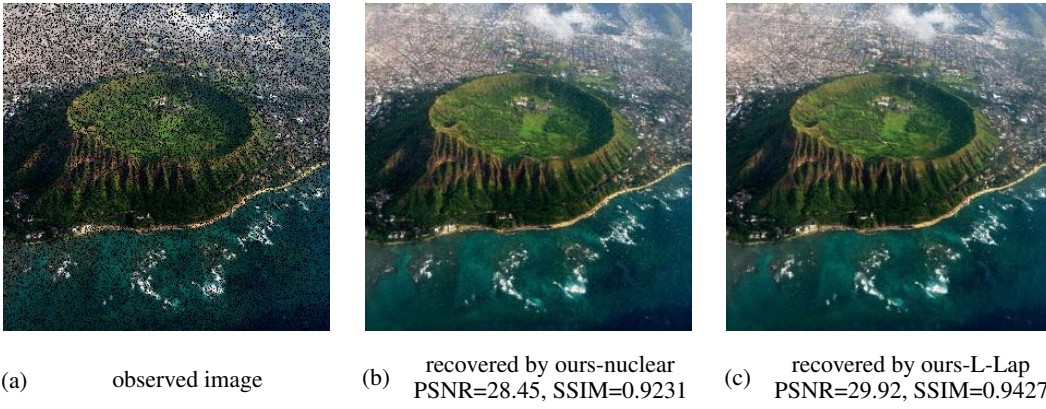

|  | | |
|---|---|---|
| (a)   observed image | (b) recovered by ours-nuclear
PSNR=28.45, SSIM=0.9231 | (c) recovered by ours-L-Lap
PSNR=29.92, SSIM=0.9427 |

Figure 6: Results of applying our method with nuclear norm and $h$-functions to an image randomly blocked by 25%.

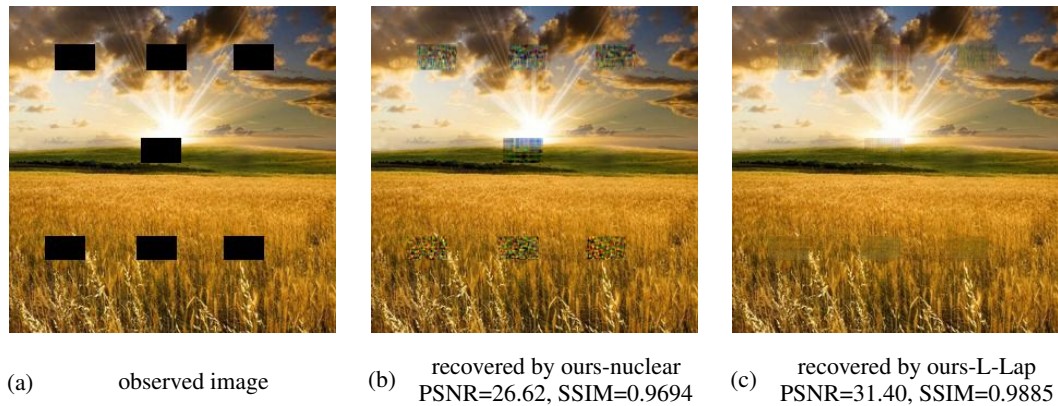

|  | | |
|---|---|---|
| (a)   observed image | (b) recovered by ours-nuclear
PSNR=26.62, SSIM=0.9694 | (c) recovered by ours-L-Lap
PSNR=31.40, SSIM=0.9885 |

Figure 7: Results of applying our method with nuclear norm and $h$-functions to an image blocked by multiple rectangular regions.

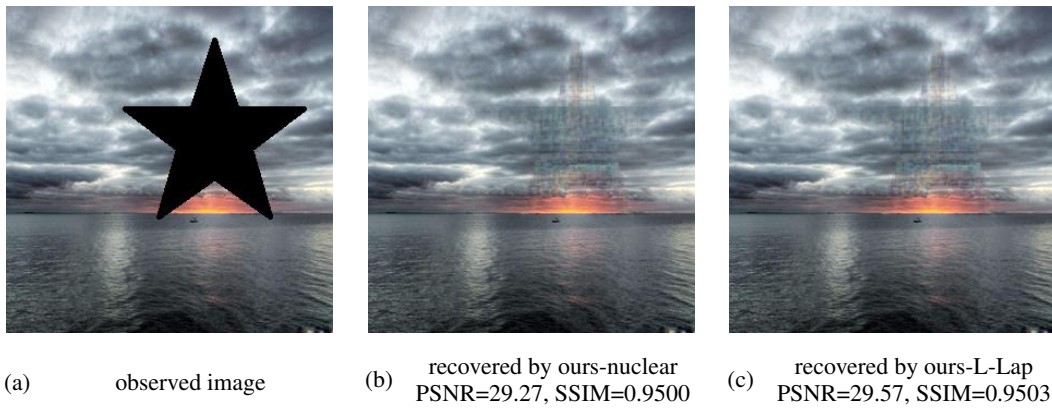

|  | | |
|---|---|---|
| (a)   observed image | (b) recovered by ours-nuclear
PSNR=29.27, SSIM=0.9500 | (c) recovered by ours-L-Lap
PSNR=29.57, SSIM=0.9503 |

Figure 8: Results of applying our method with nuclear norm and L-Lap to an image blocked by a prominent star-shaped region.

