# OpenReview forum: "Efficient Differentiable Approximation of the Generalized Low-rank Regularization"
_ICLR.cc/2024/Conference — Submitted to ICLR 2024_

### Official Review · Reviewer_Fxks · 2023-10-30

**Soundness:** 2 fair
**Presentation:** 4 excellent
**Contribution:** 2 fair
**Rating:** 5
**Confidence:** 3

**Summary:**

The authors of this paper have introduced a novel approach to address the challenges associated with low-rank regularization (LRR) in the context of various machine-learning tasks. LRR has found widespread application, but the optimization of these relaxed LRRs has typically relied on singular value decomposition, a time-consuming and non-differentiable operator that cannot be optimized using gradient-based techniques.
In response to these challenges, the authors have presented an efficient and differentiable approximation of generalized LRR. This form of LRR encompasses well-known options like the nuclear norm, the Schatten-p norm, and various nonconvex relaxations. Their proposed method allows for the seamless integration of LRR terms into loss functions, making it compatible with off-the-shelf machine-learning libraries for convenient optimization. Moreover, this approximation relies solely on matrix multiplication, a GPU-friendly operation that facilitates efficient parallel implementation.

**Strengths:**

The paper is well-written with a clear structure, making it easy for readers to follow.

**Weaknesses:**

However, it lacks some essential details, such as specifying the objective function for different problems and explaining how their algorithm derives the corresponding regularization. The conclusion is overly simplistic.

**Questions:**

In section 4.1, there are five different variations of the author's algorithm compared to other methods, which raises questions about whether there is randomness in the application of a particular method for matrix completion. Sections 4.2 and 4.3 also fall short by not providing comparisons with similar methods in the field.

---

> ### Author Response · Authors · 2023-11-17
>
> We thank the reviewer for the interest and questions to our work.
>
>
> **Question 1:**	However, it lacks some essential details, such as specifying the objective function for different problems and explaining how their algorithm derives the corresponding regularization. \
> **Response 1:**  We have presented and explained the objective function of each task at the end of the problem statement section (i.e., Section 3.1). We apologize for the unclarity, and will include a detailed discussion of the objective functions and the regularization terms in the experiment section.
>
>
> **Question 2:**	In section 4.1, there are five different variations of the author's algorithm compared to other methods, which raises questions about whether there is randomness in the application of a particular method for matrix completion .\
> **Response 2:** To address the concern about the randomness of our method’s performance, we conducted extensive experiments based on Table 1, the results are presented below. In the new experiments, each case is run for 10 times with random seeds, and we present the mean and standard deviation of the PSNR scores. Our method still consistently outperforms other baselines, and the standard deviations are also small, showing the stability of our method.
>
> | PSNR | nuclear | T-γ* | T-Lap | L-γ* | L-Lap |
> |  ----  | ----  | ----  | ----  | ----  | ----  |
> | drop 20% | 37.09±0.176 | 38.44±0.065 | 38.41±0.144 | 38.39±0.164  | 38.47±0.112 |
> | drop 30% | 35.03±0.103 | 36.24±0.125 | 36.01±0.185 |	36.19±0.132 |	36.20±0.127 |
> | drop 40% | 33.09±0.054 | 34.28±0.190 | 34.16±0.180 | 34.21±0.073 | 34.22±0.148 |
> | drop 50% | 31.39±0.087 | 32.51±0.188 | 32.31±0.145 | 32.32±0.108 | 32.38±0.090 |
> | block | 17.89±0.002 | 31.14±0.021 | 29.67±0.025 | 24.64±0.009 | 25.80±0.023 |
> | text | 26.73±0.018 | 35.07±0.059 | 34.99±0.032 | 34.97±0.053 | 34.98±0.010 |
>
> **Question 3:**	Sections 4.2 and 4.3 also fall short by not providing comparisons with similar methods in the field.\
> **Response 3:**  About video fore-background separation (Sections 4.2): The dataset used for this experiment is unannotated and without ground truth separation results. Consequently, related papers only presented qualitative results, and there is no numerical metrics can be compared with.
>
> About DNN denoising (Sections 4.3): **A key advantage of our method is that the low-rank regularization term can be conveniently applied to sophisticated DNN models, and we are unaware of any existing work has the same feature.** Most previous methods are based on ADMM, and their experiments focus on basic matrix completion tasks without DNNs involved. At this moment, we can not identify a proper baseline for fair comparison on the DNN denoising problem.

---

> > ### Comment · Reviewer_Fxks · 2023-11-22
> >
> > Thank you for your efforts in addressing my concerns during the rebuttal phase. While I appreciate your detailed responses, I must convey that I am unable to modify my opinions at this time.

---

### Official Review · Reviewer_LpEL · 2023-10-30

**Soundness:** 2 fair
**Presentation:** 3 good
**Contribution:** 3 good
**Rating:** 5
**Confidence:** 3

**Summary:**

In this paper, the authors propose a differentiable approximation of low-rank priors. The idea is to use a finite number of differentiable iterative steps to approximate the pseudo-inverse or the square root of a matrix. This differentiable approximations are used to approach low-rank regularizers in three different inverse problems. The resulting penalty function is minimized with gradient-based optimization algorithms via automatic differentiation.

**Strengths:**

- The paper is generally comprehensible, effectively written, and presents a logical flow that is easy to follow. The main contribution is clearly articulated and appropriately positioned in relation to existing literature.
- The proposed differentiable prior is novel and the idea is interesting.
- Experiments on 3 different applications prove the efficiency of the method. In particular, I find interesting the ability to stabilize the denoiser w.r.t the noise level.

**Weaknesses:**

- The optimization algorithm to minimize (1) is not explicitly presented. Although it is understood that a gradient descent or a variant is utilized, it lacks explicit clarification. Furthermore, the regularization now differentiable, but does it have Lipschitz gradient? More broadly, does the gradient descent algorithm employed to minimize (1) offer any guarantees of convergence? I believe this is a significant concern in this work, especially when mentioning that other concurrent approaches “require the loss function to be convex, which severely limits their applicability.”
- How do you compare, in performance, w.r.t computing directly (without taking care of the differentiability) the nuclear norm / Schatten-p norm with automatic differentiation ? In this case, automatic differentiation is likely to compute (approximate) subgradients.  Given the fact that the proposed method is also based on approximations, the authors should compare both approach with more details.
- There are still many specific details that are missing and that I would like to obtain (refer to the questions section).

**Questions:**

- How to you compute the power of the matrix in (6) or (7) ?
- How do you truncate the Taylor and Laguere series ?
- I would find it useful to have the precise algorithm you use in practice to compute (6) with the different approximations and the different parameters.
- Have the authors considered efficient backward pass via implicit differentiation of the fixed-point iterations ?
- In the DNN application, it is not clear if the regularization is applied during or after training.
- Do you train a different model for each sigma parameter, or do you use sigma as a parameter of the model ?
- Why the low-rank structure is only applicable to the entire image ?
- For the denoising applications, which regularizer do you use ?

---

> ### Author Response · Authors · 2023-11-17
>
> We sincerely thank the reviewer for the detailed comments and constructed feedbacks!
>
> **Question 1:** How to you compute the power of the matrix in (6) or (7) ?\
> **Response 1:** In Eq. (6) and (7), consecutive matrix powers of the form $M, M^2, …, M^p$ are required (typically $p=5$). We compute all these required matrix powers iteratively, i.e., $M_{i+1}=M_i M$ for $i=1,…,p-1$.
>
> **Question 2:** How do you truncate the Taylor and Laguere series ?\
> **Response 2:** In our experiments, the series of Taylor and Laguerre expansions are truncated up to the 5th degree. We also tested with higher degrees, but no further improvements can be observed.
>
> **Question 3:**  I would find it useful to have the precise algorithm you use in practice to compute (6) with the different approximations and the different parameters.\
> **Response 3:** Thank you for the interest in our work! Our code will be made publicly accessible on github after anonymous review.
> In our implementation, we use Mathematica to compute the required coefficients in (6) and (7), which are then input to the python code. We refer the reviewer to article [1], which shows how to compute the coefficients in the Laguerre expansions. Our code calculates coefficients of the Taylor expansion and the Laguerre expansions in a similar way. Besides, the computation is rather efficient, talking only several seconds on a personal laptop.
>
> **Question 4:** Have the authors considered efficient backward pass via implicit differentiation of the fixed-point iterations ?\
> **Response 4:** We thank the reviewer for the suggestion. Though currently our iterative approach for calculating fix-points is sufficient for practical applications, it is certainly an interesting idea to further improve its efficiency with implicit differentiation. We will leave this exploration as a future work.
>
> **Question 5:** In the DNN application, it is not clear if the regularization is applied during or after training.\
> **Response 5:** Our regularization is applied during training by adding an additional loss term to the original loss function.
>
> **Question 6:** Do you train a different model for each sigma parameter, or do you use sigma as a parameter of the model ?\
> **Response 6:** We trained a separate model for each noise level, following the previous experiment pipeline.
>
> **Question 7:** Why the low-rank structure is only applicable to the entire image?\
> **Response 7:** The application of low-rank structure in image tasks is based on the observation that, most singular values of natural images are concentrated in a few of the largest ones [2]. This phenomenon becomes less evident when the patch becomes too small and the distribution deviates from that of natural images.
>
> **Question 8:**	For the denoising applications, which regularizer do you use?\
> **Response 8:** The results reported in the paper are obtained by using the basic nuclear norm regularization (i.e., Proposition 4). We also tested with other relaxation functions, but the difference is not evident in this particular task.
>
> [1] MATHEMATICA TUTORIAL Part 2.5: Laguerre expansions (https://www.cfm.brown.edu/people/dobrush/am34/Mathematica/ch5/laguerre.html).
>
> [2] Shang, F., Cheng, J., Liu, Y., Luo, Z. Q., & Lin, Z. (2017). Bilinear factor matrix norm minimization for robust PCA: Algorithms and applications. IEEE transactions on pattern analysis and machine intelligence, 40(9), 2066-2080.

---

### Official Review · Reviewer_FgtX · 2023-10-31

**Soundness:** 4 excellent
**Presentation:** 4 excellent
**Contribution:** 4 excellent
**Rating:** 8
**Confidence:** 3

**Summary:**

The paper shows that matrix rank-function and its common surrogates can be approximated by a stochastic formulation (i.e. they can be described using means of random variables). This formulation allows approximating the rank function with a differentiable surrogate, which in turn allows the rank function to be optimized in any framework supporting gradient-descent optimization.

**Strengths:**

The method is applicable to a wide array of applications and can be plugged into any low-rank optimization/regulatization scenario.
It is fairly simple to implement.

**Weaknesses:**

It would have been nice to see some results on the approximation errors that one gets as one truncates the infinite sums in equations 6 and 7. This may help the user to choose the truncation point N.

**Questions:**

None that I can think of.

---

> ### Author Response · Authors · 2023-11-17
>
> We thank the reviewer for the interest and positive feedbacks on our work!
>
> **Question 1:** It would have been nice to see some results on the approximation errors that one gets as one truncates the infinite sums in equations 6 and 7. This may help the user to choose the truncation point N.
>
> **Response 1:** In our experiments, the series of Taylor and Laguerre expansions are truncated up to the 5th degree. We also tested with higher degrees, but no further improvements can be observed.

---

### Official Review · Reviewer_89iV · 2023-10-31

**Soundness:** 2 fair
**Presentation:** 2 fair
**Contribution:** 1 poor
**Rating:** 3
**Confidence:** 3

**Summary:**

This paper presents an effective differentiable approximation method for solving the generalized low-rank regularization problem. By integrating the differentiable regularizer into the objective function, optimization becomes an automatic process when employing gradient-based machine learning libraries. The authors apply their approach to the machine learning tasks such as text removal and foreground-background separation.

**Strengths:**

S1 The authors incorporate Ben-Israel & Cohen's iterative method for computing the matrix pseudo-inverse into their problem of minimizing low-rank regularization.

S2 The authors incorporate the Newton-Schulz iteration for matrix square root computation into their problem of minimizing low-rank regularization.

S3. The authors perform a Taylor expansion or Laguerre expansion on the smooth low-rank regularization term to render the objective function differentiable, thus enabling the use of gradient descent.

**Weaknesses:**

W1. This paper introduces a smooth approximation technique for the rank function, which is a well-explored area in the literature, featuring numerous convex and nonconvex approximation methods. However, the paper lacks an optimality analysis of the proposed technique, leaving it unclear why this particular strategy is necessary.
W2. It is unclear whether the gradient descent algorithm will converge when another sub-iteration is introduced to estimate the matrix pseudo-inverse or matrix square root.
W3.The authors suggest employing Laguerre expansion to handle the low-rank regularization function, with $\alpha_{k,p}$ representing the polynomial coefficients. The authors mention that these necessary coefficients can be computed using readily available tools like Mathematica; however, the complexity of computing these coefficients remains unclear. Furthermore, the paper does not discuss specific ranges for $k$ and $p$.

**Questions:**

See above.

---

> ### Author Response · Authors · 2023-11-17
>
> We thank the reviewer for the critical and constructive questions for our work.
>
> **Question 1:** This paper introduces a smooth approximation technique for the rank function, which is a well-explored area in the literature, featuring numerous convex and nonconvex approximation methods. However, the paper lacks an optimality analysis of the proposed technique, leaving it unclear why this particular strategy is necessary.  \
> **Response 1:** Though various convex and nonconvex application methods exist in the literature, our method possesses several unique advantages that distinguished from other works: 1) Our method can be generalized to various convex and nonconvex relaxations, while each previous work focuses on its particular form of low-rank regularization without a unified treatment. 2) Our method makes the low-rank regularization differentiable, which is compatible to modern deep learning models and libraries. 3) Our method is rather convenient to apply. The regularization term can be added in a plug-and-play fashion, and only a few lines of codes are needed to adapt to a new problem. 4) Our method is GPU-friendly and takes advantages of the efficient parallel computation, since the proposed method solely depends on matrix multiplication.
>
> Though it remains a theoretical challenge for the optimality and convergence analysis of our method, our paper presents the first low-rank approximation technique that simultaneously possess the above characteristics. Besides, the superior effectiveness and efficiency of our method have been thoroughly demonstrated in the experiments.
>
> **Question 2:** It is unclear whether the gradient descent algorithm will converge when another sub-iteration is introduced to estimate the matrix pseudo-inverse or matrix square root .\
> **Response 2:**  We would like to address the reviewer’s concern on convergence with the following points:
>
> 1.	The sub-iterations only involve several matrix multiplications/additions/subtractions, all of which are differentiable operations that permits gradient propagation. So there is no particular obstacle for gradient descent.
> 2.	Similar iterative methods for computing matrix square root have been successfully applied in previous work, while no convergence issues have been observed [].
> 3.	Empirically, our method can efficiently converge, as demonstrated in the experimental studies. Also a dedicated experiment about the convergence can be found in Appendix A.2.1.
>
> **Question 3:** The authors suggest employing Laguerre expansion to handle the low-rank regularization function, with alpha_{k,p} representing the polynomial coefficients. The authors mention that these necessary coefficients can be computed using readily available tools like Mathematica; however, the complexity of computing these coefficients remains unclear.  Furthermore, the paper does not discuss specific ranges for k and p .\
> **Response 3:** We apologize for the unclarity. Implementation details of the Laguerre expansion are presented as follows:
>
> 1.	In Mathematica, the coefficients are computed by numerically integrating a 1d function, which is a very efficient procedure.
> 2.	Particularly, it takes only several seconds to complete the computation on a personal laptop. Since the coefficients only need to be computed once, the overall computational overhead is negligible.
> 3.	The Mathematica code of our implementation will be publicly available in our repository. We refer the reviewer to article [2], which is similar to our implementation.
> 4.	In our experiments we set the ranges of k and p to be 5. In other words, the series of Taylor and Laguerre expansions are truncated up to the 5th degree. We also tested with higher degrees, but no further improvements can be observed.
>
> [1] Lin, Tsung-Yu, and Subhransu Maji. "Improved Bilinear Pooling with CNNs." Proceedings of the British Machine Vision Conference, 2017.
>
> [2] Li, Peihua, et al. "Towards faster training of global covariance pooling networks by iterative matrix square root normalization." Proceedings of the IEEE conference on computer vision and pattern recognition. 2018.
>
> [3] MATHEMATICA TUTORIAL Part 2.5: Laguerre expansions (https://www.cfm.brown.edu/people/dobrush/am34/Mathematica/ch5/laguerre.html).
>
> We hope our responses would address the reviewer’s concerns and clarify the misunderstandings. We also welcome the reviewer for any further questions and suggestions for our work.

---

### Official Review · Reviewer_XkwY · 2023-10-31

**Soundness:** 2 fair
**Presentation:** 2 fair
**Contribution:** 3 good
**Rating:** 5
**Confidence:** 5

**Summary:**

This paper introduces a very interesting ideal to represent the nuclear norm as well as non-convex low rank regularizes as Expectation of differentiable function, and use the differentiable low-rank regularization model for matrix completion and video foreground-background separation.

**Strengths:**

The results presented in this paper is novel and significant, especially for the equivalent expressions of nuclear norm and the non-convex singular value regularization functions.

**Weaknesses:**

1.	The writing of this paper is crude. The authors should present the detailed algorithms as well as convergence analysis when using the proposed model for specific applications such as matrix completion as well as video foreground and background separation.
2.	Some important related works such as [1-5] should be discussed and compared in this work as they are representative works that are dealing with the same problem as the current work.
3.	Some of the presentation is not correct, such as “…key problem of the matrix factorization is that it demands strong prior knowledge on the matrix rank…”. To my knowledge, the variational representation of the nuclear norm as well as the non-convex regularization do not need to know the exact rank, such as [1,2,4,5].
4.	The authors argue that the SVD is not differentiable, please refer to [6] for more information.
5.	Some of the experimental results are vague. Specifically, in Table 1, the running time of MSS is slower than IRNN. As for as I know, MSS is a representative factorization-based model which is much faster than the IRNN. Besides, the proposed model provides an equivalent representation of the nuclear norm and non-convex rank regularization function, the overall mathematic model is the same as the low rank regularization function implied on the original full matrix. Why does the performance of the proposed method present in Table 1 is much better than existing methods with the same mathematical model? Where does the performance gain come from?
[1] Ornhag, M. V., Iglesias, J. P., & Olsson, C. (2021). Bilinear parameterization for non-separable singular value penalties. In Proceedings of the IEEE/CVF Conference on Computer Vision and Pattern Recognition (pp. 3897-3906).
[2] Ornhag, M. V., Olsson, C., & Heyden, A. (2020). Bilinear parameterization for differentiable rank-regularization. In Proceedings of the IEEE/CVF Conference on Computer Vision and Pattern Recognition Workshops (pp. 346-347).
[3] Yao, Q., Kwok, J. T., Wang, T., & Liu, T. Y. (2018). Large-scale low-rank matrix learning with nonconvex regularizers. IEEE transactions on pattern analysis and machine intelligence, 41(11), 2628-2643.
[4] Jia, X., Feng, X., Wang, W., & Zhang, L. (2020). Generalized unitarily invariant gauge regularization for fast low-rank matrix recovery. IEEE Transactions on Neural Networks and Learning Systems, 32(4), 1627-1641.
[5] Shang, F., Cheng, J., Liu, Y., Luo, Z. Q., & Lin, Z. (2017). Bilinear factor matrix norm minimization for robust PCA: Algorithms and applications. IEEE transactions on pattern analysis and machine intelligence, 40(9), 2066-2080.
[6] Papadopoulo, T., & Lourakis, M. I. (2000). Estimating the jacobian of the singular value decomposition: Theory and applications. In Computer Vision-ECCV 2000: 6th European Conference on Computer Vision Dublin, Ireland, June 26–July 1, 2000 Proceedings, Part I 6 (pp. 554-570). Springer Berlin Heidelberg.

**Questions:**

Please refer to the weaknesses part.

---

> ### Author Response · Authors · 2023-11-17
>
> Thanks for your hard work and valuable comments of this work. Here are responses for your concerns and questions:
>
> **Question 1:** The authors should present the detailed algorithms  as well as convergence analysis  when using the proposed model for specific applications such as matrix completion as well as video foreground and background separation.\
> **Response 1:** We thank the reviewer for the suggestions. We will add a detailed algorithm description in the Appendix in the revised version.
> Rigorous theoretical convergence analysis is challenging in our case. Since our method novely applies several approximations for computing matrix root and inversion, existing analysis techniques cannot be directly applied. Nevertheless, our method converges quite well in practice. One representative example is the convergence result shown in Appendix A.2.1.
>
> **Question 2:** Some important related works such as [1-5] should be discussed and compared in this work as they are representative works  that are dealing with the same problem as the current work.\
> **Response 2:** Thank you for pointing out these important related works. In the table below we listed some key aspects of these researches and make comparisons with our work. Particularly, we would like to point out that:
>
> 1.	Our method is based on a stochastic approximation of the low-rank regularization function, which to the best of our knowledge, is a completely novel approach compared with other related work.
> 2.	Since our optimization method is gradient-based, it can be convenient applied to popular deep learning models and libraries.
> A discussion and comparison with these related works will be included in the revised manuscript.
>
> | Related Work | Low-rank Form | Key technique | Optimization Method |
> |---|---|---|---|
> |Ornhag, 2021 [1]| $\sum h(\sigma_i)$, any smooth $h$| Bilinear matrix factorization, Taylor expansion and quadratic approximate $h$| ADMM|
> |Ornhag, 2020 [2]| $\sum a_i \sigma_i+b_i$| Bilinear matrix factorization | ADMM|
> |Yao, 2018 [3]| $\sum h(\sigma_i)$, $h$ is concave| Approximate proximal operator with power method | Proximal algorithm |
> |Jia, 2020 [4]| $\sum h(\sigma_i)$, $h$ is concave| Bilinear matrix factorization | ADMM|
> |Shang, 2017 [5]| Schatten-1/2 and 2/3 quasi-norms| Bilinear matrix factorization| ADMM|
> |Ours| $\sum h(\sigma_i)$, any smooth $h$| Stochastic approximation | Gradient descent |
>
> **Question 3:** 	To my knowledge, the variational representation of the nuclear norm as well as the non-convex regularization do not need to know the exact rank, such as [1,2,4,5] .\
> **Response 3:** Though these methods do not need to know the exact rank, they need to know “the upper bound of the exact rank”: most factorization methods assume $A=BC$ where $A\in R^{m\times n}$, $B\in R^{m\times k}$, and $C\in R^{k\times n}$. Here the choice of $k$ requires strong prior knowledge, and many previous works show that an inappropriate choice will deteriorate the performance.
>
> **Question 4:**  The authors argue that the SVD is not differentiable, please refer to [6] for more information.\
> **Response 4:** We sincerely thank the reviewer for pointing out the result unknown to us before. Indeed, it presents a method for computing the gradients of SVD. The related discussions in our manuscript will be modified accordingly. However, though the result shows that differentiating SVD is possible, it is far from being practical. The prohibitive $O(N^4)$ complexity for computing gradients is unacceptable in our setting.
>
> **Question 5:**	**(About running time)** As for as I know, MSS is a representative factorization-based model which is much faster than the IRNN .  **(About performance gain)** Where does the performance gain come from?\
> **Response 5:**
>
> About running time: For all the compared baselines, we use the codes from public repositories and locally evaluated their performance with the same environment. Particularly, the code of MSS is taken from [6], and the code of IRNN is from [7]. The exact running times may vary between implementations.
>
> About performance gain: There are two major reasons that explains the superior performance of our method:
>
> 1.	Our method benefits from the non-convex relaxation of the rank function. Many of the compared baselines use the nuclear norm or Schatten-p quasi-norms with p<1, while our method utilizes more sophisticated relaxation.
> 2.	Even the non-convex relaxation function and the mathematical models are the same, different methods have unequal convergence behaviors. We also note that while many related works have presented convergence analysis, these results typically characterize the worst-case performance, and may not fully explain the empirical performance.
>
> [6] https://github.com/xuchen314/MSS/tree/master
>
> [7] https://github.com/canyilu/Iteratively-Reweighted-Nuclear-Norm-Minimization

---

### Public Comment · ~Mingyu_Kim2 · 2023-12-05
**Comment by Mingyu Kim**

I appreciate for authors to dedicate low-rank regularization.
I'd like to ask you to open source codes for validation of experimental results on Matrix completion for image pinpointing and regularizing DNN-based denoising models.

According to your manuscript, the experimental results look promising, but I could not verify them myself.
I would be thrilled to understand and appreciate the impact of this work, if possible.

---

### Meta-Review · Area_Chair_eS1W · 2023-12-05

**Metareview:**

The paper presents a novel approach for differentiable approximation of low-rank regularizers, applicable for instance to matrix completion. While the concept is interesting, addressing a significant challenge in low-rank regularization, the execution has notable weaknesses highlighted by all the referees. The paper lacks detailed algorithmic presentations, convergence analysis (how does the tradeoff between accuracy and differentiability works?), and comprehensive experimental validations. For these reasons, I recommend to reject it from ICLR'23.

**Justification For Why Not Higher Score:**

lacks detailed algorithmic presentations and convergence analysis, omits essential comparisons with related works, and contains inaccuracies (e.g. SVD differentiability, ) and vague experimental results.

**Justification For Why Not Lower Score:**

N/A

---

### Decision · Program_Chairs · 2024-01-16

Reject